# Prevalence and factors associated with adverse drug reactions among patients on highly active antiretroviral therapy at a tertiary hospital in south western Uganda: A cross-sectional study

**Nangosya Moses**[1], **Sarad Pawar Naik Bukke**[2], **Narayana Goruntla**[3], **Daniel Chans Mwandah**[4,5], **Bontu Aschale Abebe**[6,7], **Fredrick Atwiine**[1], **Muyindike Rhoda Winnie**[8], **Tadele Mekuriya Yadesa**[1,3]*

1 Department of Pharmacy, Mbarara University of Science and Technology, Mbarara, Uganda,
2 Department of Pharmaceutics and Pharmaceutical Technology, School of Pharmacy, Kampala International University, Ishaka, Uganda, 3 Department of Clinical Pharmacy and Pharmacy Practice, School of Pharmacy, Kampala International University, Ishaka, Uganda, 4 Department of Pharmacology and Therapeutics, Mbarara University of Science and Technology, Mbarara, Uganda, 5 Department of Pharmacology and Toxicology, School of Pharmacy, Kampala International University, Ishaka, Uganda, 6 School of Nursing, Kampala International University, Ishaka, Uganda, 7 Department of Midwifery, Ambo University, Ambo, Ethiopia, 8 Department of Internal Medicine, Mbarara University of Science and Technology, Mbarara, Uganda

* maatiikoo4@gmail.com

## Abstract

### Background

HIV/AIDS remains a global health challenge, with significant prevalence in sub-Saharan Africa. Highly active antiretroviral therapy (HAART) is the mainstay treatment for HIV, and the number of people living with HIV (PLWHIV) on HAART has considerably increased worldwide. The use of HAART has led to improved patient outcomes; however, it is associated with adverse drug reactions (ADRs) and drug-drug interactions (DDIs), which pose serious concerns in the management of patients with HIV. The aim of the study was to determine the prevalence and factors associated with ADRs among patients on HAART.

### Methodology

This was a hospital-based cross-sectional study carried out among 312 HIV patients on HAART attending HIV clinics at Mbarara Regional Hospital. Data was collected using an interviewer-administered, semi-structured questionnaire and a review of patient charts. ADRs were assessed for causality and categorized using Naranjo ADR assessment scale into probable, possible and definite, for severity using the modified Hartwig and Siegel criteria into mild, moderate and Severe, and for preventability using Schumock and Thornton criteria into definite, probable and non-preventable. Lexicomp® Drug Interaction Checker software was used to identify and rate clinically significant drug-drug interactions.

**Data availability statement:** All relevant data are within the manuscript and its Supporting information files.

**Funding:** The author(s) received no specific funding for this work.

**Competing interests:** The authors have declared that no competing interests exist.

**Abbreviations:** ABC, Abacavir; ADEs, Adverse Drug Events; ADRs, Adverse drug reactions; AIDS, Acquired Immunodeficiency Syndrome; BNF, British National Formulary; ART, Antiretroviral therapy; ARVs, Antiretrovirals; ATV/r, Atazanavir/ritonavir; AZT, Zidovudine; CD4, Clusters of differentiation; COR, Crude odds ratios; DTG, Dolutegravir; EFV, Efavirenz; HAART, Highly Active Anti-Retroviral Treatment; HIV, Human immuno deficiency virus; EMTCT, Eliminating mother to child transmission of HIV; IRB, Institutional Review Board; ISS, Immune Suppression Syndrome; LPV/r, Lopinavir/ritonavir; MoH, Ministry of Health; MRRH, Mbarara Regional Referral Hospital; MUST, Mbarara University of Science and Technology; NVP, Nevirapine; OTC, Over the counter; PLWHA, People Living with HIV/AIDS; REC, Research Ethics Committee; TAF, Tenofovir alafenamide; TDF, Tenofovir; 3TC, Lamivudine; SSA, Sub Saharan Africa; UNAIDS, United Nations Programme on HIV/AIDS; WHO, World Health Organization

The prevalence of ADRs and potential DDI was analyzed using descriptive statistics while logistic regression analysis was used to establish the association of variables.

## Results

312 patients were interviewed and their records reviewed. The prevalence of ADRs during this study was 76.0%. On assessment, 78.3% of the ADRs were mild and 76.6% of ADRs were definitely preventable. CD4 count below 200 cells/mm$^3$ (AOR = 1.00, 95% CI: 1.00–1.02; $p$ value = 0.04), primary education level (AOR = 3.27, 95% CI: 1.34–7.95; $p$ value = 0.009), and secondary education level (AOR = 3.64, 95% CI: 1.39–9.52; $p$ value = 0.009) were identified as independent risk factors. Patients who experienced a significant DDI were 5.66 times more likely to experience an ADR ($p$ value = 0.02, 95% CI: 1.32–24.18).

## Conclusion

There is a high prevalence of adverse drug reactions among patients with HIV on HAART. Low CD4 count and lower education levels are risk factors for ADRs in this population; therefore, tailored interventions to these subgroups should be implemented for early ADR identification and management. Significant drug-drug interactions are highly associated with the occurrence of ADRs among HIV patients on HAART, which calls for intensified pharmacovigilance and pharmaceutical care in this population.

## Background

Globally, about 39.9 million persons live with HIV [1], of which 30.7 million people had access to antiretroviral therapy (ART) by the end of 2023 [2]. Sub-Saharan Africa (SSA) accounts for about 67% of all people living with HIV worldwide [3]. Uganda, which is one of the ten high-burden countries in SSA, had a nationwide HIV prevalence of 5.1% among adults aged 15–49 years in 2023 [3], with the Mbarara district having a prevalence of 13.1% [4,5]. The number of people gaining access to highly active antiretroviral therapy (HAART) is increasing globally. HAART involves use of a combination of three or more antiretroviral drugs to effectively suppress HIV replication [6]. This approach aims to restore immune function, halt disease progression, improve survival rates, reduce morbidity, and enhance the overall quality of life for those with HIV [7]. The use of HAART has led to a remarkable decrease in AIDS-related mortality by 51% from about 1.3 million deaths in 2010–630,000 in 2023 [2,8].

Despite the effectiveness of HAART in managing HIV, there is a high risk of adverse drug reactions (ADRs) associated with HAART due to the complexity, toxicity, and potential for drug-drug interactions [9,10]. Additionally, these patients tend to have comorbid conditions that also require drug therapy [11], which predisposes these patients to ADRs from the associated use of drugs. An adverse drug reaction can be defined as a harmful or unpleasant reaction that is due to an intervention that is related to the use of a medicinal product that predicts hazard from future administration and hence warrants prevention, treatment, or adjustment in dosage regimen, or discontinuation of the product [12,13]. ADRs negatively impact patients' quality of life, lead to treatment discontinuation, medication non-adherence, and poor patient outcomes [14–16].

Healthcare professionals play a key role in preventing, identifying, and managing ADRs by closely monitoring patients, promoting medication safety practices, and providing appropriate education and counseling regarding potential risks and side effects [17–19]. Pharmacists in

particular can offer comprehensive strategies that focus on medication reconciliation, proper dosing, and patient education, which help minimize ADRs and improve patient outcomes [20,21]. Several studies have been carried out in patient populations on specific HIV treatment regimens with varying prevalence and factors leading to ADR occurrence in this population in both time and location [22–24].

Given the benefits of HAART in managing HIV and the potential impact of ADRs on patient outcomes, it is key to understand the prevalence and factors associating with ADR occurrence in patients on HAART. However, currently there is a lack of comprehensive research and evidence-based results regarding the prevalence and factors contributing to ADRs, especially among HIV patients on HAART in our setting. The study therefore aimed to determine the prevalence and factors associated with ADR occurrence among patients on HAART at a tertiary hospital in Uganda.

## Methods

### Study setting and period

The Mbarara Regional Referral Hospital's (MRRH) HIV/Immune Suppression Clinic (ISS), located 280 km from Kampala, served as the study's location. MRRH is a 350-bed regional referral hospital located in southern Uganda. The hospital is currently serving a population of about four million people in the catchment area of the districts of Kiruhura, Mbarara, Sheema, Bushenyi, Rwampara, Ibanda, Buhweju, Rubirizi, Mitooma, Isingiro, Lyantonde, Rakai, Ntungamo, and other neighbouring districts. The HIV clinic has a total of 11,218 active clients on ART and receives about 80–150 clients per day. A physician leads the clinic as the in-charge, supported by a paediatrician, medical officers, a pharmacist, 5 registered nurses, 8 enrolled nurses, 2 dispensers, 4 laboratory technicians, 4 counsellors, a biostatistician, 4 data clerks, and other support staff. The clinic has a separate laboratory embedded within it for common screening tests, as well as some routine follow-up and monitoring parameters. We conducted the study between 11th September 2023 and 31st of October 2023.

### Study design

This was a prospective cross-sectional study conducted among HIV patients actively on HAART who attended ISS clinic of the Mbarara Regional Referral Hospital.

### Study population

The study included all adult PLW HIV (18 years and older) who were on HAART at the ISS Clinic of MRRH during the study period, actively on follow-up, and willing to participate. We excluded patients who started HAART less than a month ago and those with incomplete records in their files (HAART regimen, start date, weight) from this study.

### Sample size determination

The sample size of this study was estimated using Fisher's formula for estimating sample size, $\mathbf{n} = \mathbf{Z}^2 \alpha /_2 \, \mathbf{P}(1 - \mathbf{P}) / \delta^2$ [25]. The prevalence of ART-related ADRs of 24.4% was obtained from a systemic review study in Africa [26]. With the addition of 10% to compensate for incomplete data, the sample size was determined to be **312 patients.**

### Sampling technique

At the HIV clinic, we assigned consecutive numbers to patients' files every day of the study period, starting from one, based on their follow-up schedules. We generated random numbers

using MS-Excel version 2016. Every patient at the clinic received an equal chance of selection. The same procedure continued on clinic days until a number of 312 was achieved, which was our sample size.

## Data collection tool and procedure

We obtained informed consent from each patient to participate in the study. We used a data abstraction form to extract data from patients' medical charts, including diagnostic data. We conducted interviews with patients using a structured interviewer-administered questionnaire to gather data not available in the patient files. We evaluated the known adverse reaction profile of each drug using Ugandan Clinical Guidelines 2023 [27] and British National Formulary (BNP) version 85. An adverse drug reaction was defined as any untoward reaction to a medication [28]. We used the Naranjo ADR assessment scale [29] to rate the causal relationship between an ADR and the suspected medication. We excluded all doubtful ADRs and only considered those rated as possible, probable, or definite for discussion and verification. We established the severity of ADRs by applying the modified Hartwig and Siegel criteria, which consisted of seven items and three severity categories: "mild, moderate, and severe ADRs" [30]. ADRs were further assessed for preventability by using the 9-item Schumock and Thornton criteria, categorizing adverse drug reactions into definitely preventable, probably preventable, or not preventable [31]. Lexicomp® Drug Interaction Checker software was utilized to identify clinically significant drug-drug interactions (DDIs) and classified them according to risk rating: A (no known interaction), B (no action needed), C (monitor therapy), D (consider therapy modification), and X (avoid combination).

Research assistants followed the Ugandan Ministry of Health's Covid-19 and Ebola Protocols. These included use of personal protective equipment like double masking, sanitizing before, during, and after handling patient files, disinfecting the area, and hand washing with soap and water.

## Data quality control

Three research assistants were recruited and trained on data collection procedures. The data collection tool was pretested with 20 participants to check for any ambiguity, reproducibility and appropriateness. Good clinical practice guidelines and a well-defined data collection procedure were followed. The principal investigator closely supervised the research assistants and provided timely revisions about data completeness and appropriateness.

## Data analysis

Data was checked for completeness, entered into Microsoft Excel 2016 for data cleaning and then exported to Stata version 13 for analysis. The prevalence of ADRs and potential DDI was analyzed using descriptive statistics and presented in measures of central tendency, frequencies, and percentages. Bivariate and multivariate logistic regression was used to identify the factors associated with ADRs. The multivariate logistic regression included variables that had a p-value of less than 0.25 in the bivariate logistic regression. A p-value of <0.05 and a 95% confidence interval were used as cutoff points for determining the statistical significance of associations.

## Ethical approval and consent to participate

This study was conducted in accordance with the Declaration of Helsinki. The study was approved by the Mbarara University of Science and Technology, Research and Ethics Committee (Reference Number: MUST-2023–1074). Clearance to carry out the study was obtained from the Mbarara Regional Referral Hospital. As all the participants were 18 years of age or

older, written informed consent was obtained directly from each participant before recruiting in this study. Patients were given a choice between English or Runyankore versions of the approved consent forms. For patients who could not read, the patient's caretaker or any other person approved by the patient read the consent to the patient and served as a witness.

## Results

### Socio-demographic characteristics of participants

A total of 312 patients were interviewed and their records reviewed. The majority of the participants were within the age range of 45–55 years (33.0%). There were more females (60.9%) than males (39.1%). The age range was 23–80 years with a mean of 45.2 (SD, 10.6). Most participants were married (58.7%), those who had primary education were 49.4%, peasants were 36% and 53.5% lived in an urban area (Table 1).

### Clinical characteristics at the beginning of ART

Among the selected patients, more than 99% had started HAART at WHO stage 1. Over 50% of patients began HAART with a CD4 count greater than 200 cells/mm³ (Table 2). 30.8% of the patients had other comorbidities, with hypertension being the most common (Fig 1).

**Table 1. Socio-demographic characteristics of participants with HIV on HAART at Mbarara Regional Referral Hospital.**

| Variable | Category | Frequency n=312 | Percentage |
|---|---|---|---|
| **Age group (years)** | <35 | 54 | 17.3 |
| | 35-<45 | 91 | 29.2 |
| | 45-<55 | 103 | 33.0 |
| | ≥55 | 64 | 20.5 |
| **Sex** | Male | 122 | 39.1 |
| | Female | 190 | 60.9 |
| **Marital Status** | Single | 18 | 5.8 |
| | Married | 183 | 58.7 |
| | Divorced | 51 | 16.4 |
| | Widowed | 60 | 19.2 |
| **Educational attainment** | None | 34 | 10.9 |
| | Primary | 154 | 49.4 |
| | Secondary | 90 | 28.9 |
| | Tertiary | 34 | 10.9 |
| **Occupational status** | Unemployed | 9 | 2.9 |
| | Employed | 37 | 11.9 |
| | Peasant | 112 | 36.0 |
| | Business | 82 | 26.3 |
| | Others | 72 | 23.1 |
| **Place of residence** | Rural | 90 | 28.9 |
| | Semi urban | 55 | 17.6 |
| | Urban | 167 | 53.5 |
| **Alcohol Use** | Yes | 98 | 31.4 |
| | No | 214 | 68.6 |
| **Smoking** | Yes | 28 | 9.0 |
| | No | 383 | 91 |

**Table 2.  Participants clinical characteristics patients with HIV on HAART at MRRH.**

| Variable | Category | Frequency | Percentage |
|---|---|---|---|
| Baseline CD4 count | >200 | 189 | 61.6 |
| | <200 | 118 | 38.4 |
| Current CD4 count | >200 | 256 | 83.9 |
| | <200 | 49 | 16.1 |
| Comorbidity | Yes | 93 | 30.8 |
| | No | 209 | 69.2 |
| Opportunistic Infection | Yes | 118 | 38.2 |
| | No | 191 | 61.8 |
| Medication Use Counselling | Yes | 311 | 99.7 |
| | No | 1 | 0.3 |

**Others** - Heart failure, kidney disease, liver disease, schizophrenia, cancer, asthma, intrinsic lung disease, ischemic heart disease, osteoarthritis, anemia, thrombocytopenia, sickle cell disease, hepatitis B, hepatitis C, tuberculosis.

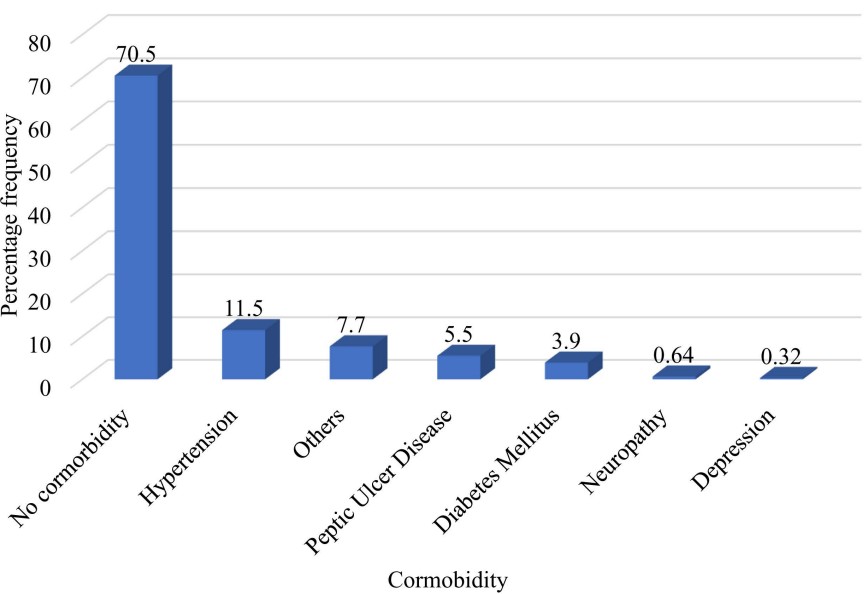

**Fig 1.  Comorbidity status of patients with HIV on HAART at MRRH.**

## HAART drug regimens

The initial regimen was TDF/3TC/EFV for 153 (49.4%), TDF/3TC/DTG for 45 (14.1%), AZT/3TC/NVP for 50 (16.0%), AZT/3TC/EFV for 21 (6.7%), TDF/3TC/LPV/r for 12 (3.9%), TDF/3TC/ATV/r for 9 (2.9%) and other ART 14 (4.5%) of patients. The majority of the patients were currently on TDF/3TC/DTG 292 (92.3%), followed by TDF/3TC/EFV (2.6%) and 1.9% were on ABC/3TC/DTG (Table 3). About 185 (59.5%) patients reported using other medicines other than for HIV treatment. From those who took other drugs other than ART, 16 (5.1%) patients were on co-trimoxazole and 305 (98.1%) patients had ever been on isoniazid prophylaxis. About 75 (24.7%) of the patients reported using herbal drugs.

**Table 3. Current HAART regimen among Patients with HIV at MRRH.**

| HAART Regimen | Frequency | Percentage |
|---|---|---|
| TDF/3TC/DTG | 292 | 92.3 |
| TDF/3TC/EFV | 8 | 2.6 |
| ABC/3TC/DTG | 6 | 1.9 |
| AZT/3TC/DTG | 1 | 0.3 |
| AZT/3TC/NVP | 1 | 0.3 |
| TDF/3TC/ATZ/r | 2 | 0.64 |
| TDF/3TC/LPV/r | 2 | 0.64 |

**\* TDF** (Tenofovir Fumarate), **3TC** (Lamivudine), **DTG** (Dolutegravir), **EFV** (Efavirenz), **ABC** (Abacavir), **AZT** (Zidovudine), **NVP** (Nevirapine), **ATZ/r** (Atazanavir/ritonavir), **LPV/r** (Lopinavir/ritonavir).

## Prevalence of ADR

The number of patients who reported at least one ADR was 239 out of 312, representing 76.6% (Fig 2).

## ADR assessment, severity and preventability

Using the Naranjo ADR causality rating scale, 60.7% of ADRs were rated as probable, 39% as possible and 0.3% rated as definite. Modified Hartwig and Siegel's scale was used for severity assessment, where 78.3% of the ADRs were mild, 21.4% moderate and 0.3% severe. Preventability assessment was done by using the Schumock and Thornton scale and it was found that 76.6% of the ADRs were definitely preventable, 11.0% were probably preventable, and 12.4% were non-preventable (Fig 3).

Out of the 76% ADRs in HAART detected, TDF was the most implicated with 141 (47%), DTG with 78 (26%), 3TC 71 (23.8%) and others 10 (3.2%) of the ADRs. The most commonly reported adverse drug reaction was low back pain (17.2%), difficulty falling asleep (9.9%), fatigue 66 (9.7%), skin manifestations (9.3%) and GIT manifestations (8.2%). Common actions taken by those reported were counselling about the drug and encouraging them to continue with the drugs or withholding the drugs (Table 4).

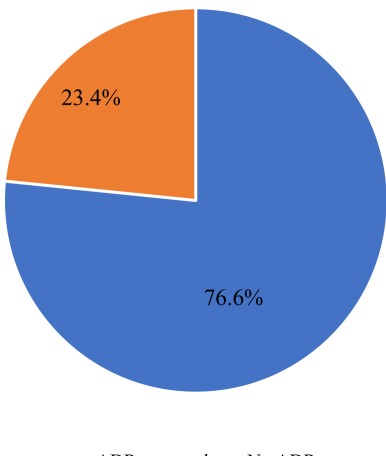

**Fig 2. Prevalence of ADRs among patients living with HIV at MRRH.**

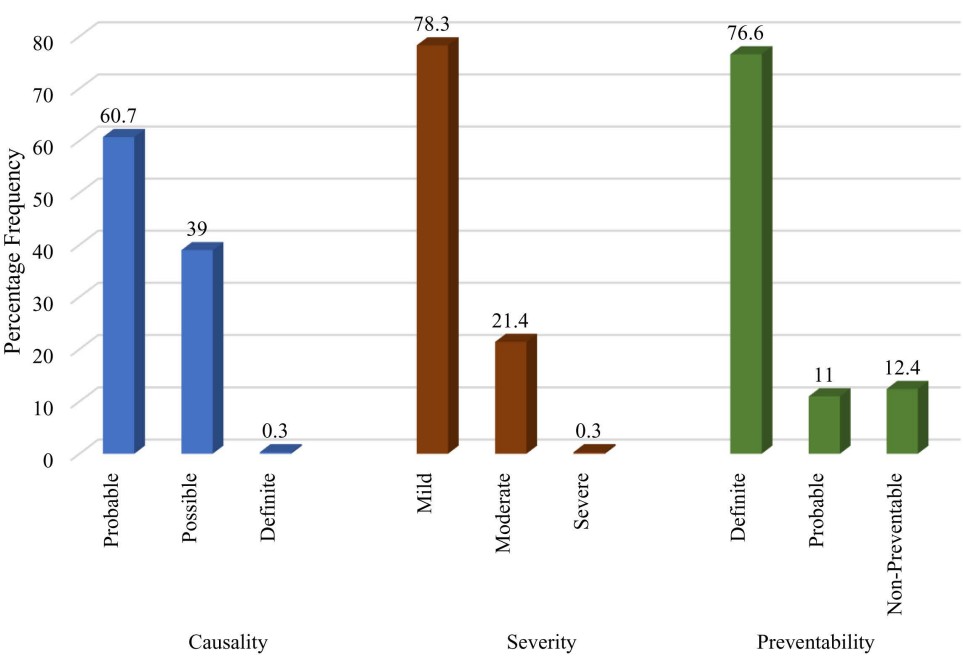

**Fig 3. ADR assessment, severity and preventability among patients with HIV on HAART at MRRH.**

## Factors associated with adverse drug reactions

Variables with *p*-value less than 0.25 at bivariate analysis included polypharmacy, age, gender, education level, baseline CD4 count, current CD4 count, comorbidity, change of regimen, over-the-counter medicine use (OTC), and herbal drug use. These variables were included in multivariable logistic regression and only current CD4 count below 200 cells/mm$^3$ (AOR = 1.88, 95% CI: 1.23–1.99; *p* value = 0.041) vs ≥200 cells/mm$^3$, and primary level of education (AOR = 3.27, 95% CI: 1.34–7.95; *p* value = 0.009) and secondary level of education (AOR = 3.64, 95% CI: 1.39–9.52; *p* value = 0.007) vs tertiary educational level, were significantly associated with occurrence of at least one ADR (Table 5).

## Discussion

In this study, the prevalence of ADR occurrence among HIV patients on HAART was determined to be 76.6%. The prevalence is considerably higher than the one reported from pooled studies in Africa, which gave a prevalence of 28% [26], and also higher than 7.9% reported in Nigeria [32], 25.5% in Ethiopia [33], 13.8% in India [34] and 9.4% reported in Ghana [35]. The possible explanation could be due to differences in the pharmacovigilance practices in different settings, the study employed in some studies, the number of participants in the study and the duration of follow-up period of different studies. Given the ever-changing HIV treatment regimen [36,37], the difference could be as a result of the difference in regimen studied in different studies, as ADR occurrence has shown to differ based on the toxicity of the regimen used [38,39]. However, the reported prevalence in this study is lower than 85.5% reported in Brazil [40] and 94% from Iran [41]. Possible reasons for a lower prevalence could be the cross-sectional design in the current study, which typically captures data at a single point in time, while prospective studies, like the study from Brazil, are generally more robust in terms of data collection and follow-up, which may lead to a more comprehensive assessment of ADRs. The high prevalence of ADRs in this study highlights a significant public

**Table 4. Commonly Reported ADRs among patients with HIV on HAART at MRRH.**

| Category of ADR | Frequency (N=657) | Percentage (%) |
|---|---|---|
| Low back pain | 117 | 17.8 |
| Difficulty falling asleep | 67 | 10.2 |
| Fatigue | 66 | 10.0 |
| Skin manifestations (Rashes & itching) | 63 | 9.6 |
| GIT manifestations (Nausea, Vomiting and diarrhea) | 56 | 8.5 |
| Abdominal pain | 55 | 8.4 |
| Drowsiness/Dizziness | 43 | 6.5 |
| Excessive eating | 30 | 4.6 |
| Low appetite | 27 | 4.1 |
| Increased thirst | 26 | 4.0 |
| Headache | 17 | 2.6 |
| Numbness of the extremities | 16 | 2.4 |
| Renal problems | 8 | 1.2 |
| Nightmares | 6 | 0.9 |
| Bone pains | 6 | 0.9 |
| Chest pain | 5 | 0.8 |
| Right upper quadrant pain | 5 | 0.8 |
| Difficulty breathing | 4 | 0.6 |
| Eye itching | 4 | 0.6 |
| Cardiovascular problems | 4 | 0.6 |
| Joint pain | 4 | 0.6 |
| Muscle cramps | 4 | 0.6 |
| Hyperglycemia | 4 | 0.6 |
| Others* | 20 | 3.1 |

*constipation, excess urination, abnormal discharge, CNS disturbances (hallucinations, depression), cough, low sex drive, dementia, breast tenderness, dyspnea, hearing defects/tinnitus, dehydration, stomach fullness, limb swelling, over sleeping, yellow urine, tenesmus.

**Difficulty falling asleep** - dissatisfaction with sleep quality or quantity; difficulty initiating sleep, difficulty maintaining sleep, the sleep difficulty occurs at least three times a week. **Low appetite** – reduced desire to eat food. **Excessive eating and drinking** – Feeling of hunger and thirst leading often eating or drinking much larger than usual amounts of food or drinks at a time or over short period of time. **Renal problems** – Acute or chronic kidney injury based on serum creatinine levels and glomerular filtration rate. **Cardiovascular problems** - Increased serum creatine kinase or hypotension, or palpitations.

health concern. For practice, healthcare providers should prioritize enhanced pharmacovigilance and thorough patient monitoring to better identify and manage ADRs. ADR reporting systems should be strengthened and standardized and ensure that they are implemented uniformly at all healthcare levels.

In our study, back pain, skin manifestations including rash and itching, and GIT manifestations were the most reported ADRs. Back pain due to antiretroviral drugs has been linked to neuropathy caused by some of these drugs, although HIV as a chronic illness has also may cause back pain due to increased inflammation, exaggerated immune system responses, and nerve damage, co-existing conditions [42,43]. Some drugs, such as Non-Nucleoside Reverse Transcriptase Inhibitors such as efavirenz, have shown to cause rash and itching and the profound increase in CD4 counts during the initial phase of treatment has been proposed as a possible explanation for the development of the drug-induced skin rash [44]. The gastro-intestinal symptoms, including diarrhea, nausea, vomiting, and dyspepsia, is caused by GI intolerance by most ARV drug combinations, which usually appears during the first weeks

**Table 5. Univariate and multivariate logistic regression of the factors associated with ADR among patients living with HIV at MRRH.**

| Variables | Category | COR (95 CI) | P value | AOR (95 CI) | P value |
|---|---|---|---|---|---|
| Gender | Male | 1 | | | |
| | Female | 1.29 (0.76-2.20) | 0.910 | | |
| Age (years) | < 35 | 1 | | | |
| | 35-55 | 0.98 (0.45-2.10) | 0.953 | | |
| | >55 | 1.69 (0.69-4.11) | 0.250 | | |
| Polypharmacy | No | 1 | | 1 | |
| | Yes | 2.22(1.07-4.61) | 0.032 | 1.40(0.98-1.88) | 0.241 |
| Education | Tertiary | 1 | | 1 | |
| | Primary | 3.26(1.49-7.16) | 0.003 | 3.27(1.34-7.95) | **0.009** |
| | Secondary | 3.65(1.54-8.68) | 0.003 | 3.64(1.39-9.52) | **0.007** |
| Comorbidity | No | 1 | | 1 | |
| | Yes | 2.36(1.22- 4.57) | 0.011 | 1.55(0.13- 1.04) | 0.098 |
| Current CD4 | > 200 | 1 | | 1 | |
| | < 200 | 4.21 (2.34-8.01) | 0.008 | 1.88(1.23-1.99) | **0.041** |
| Baseline CD4 | > 200 | 1 | | 1 | |
| | < 200 | 1.12 (1.01-1.82) | 0.029 | 1.01(0.87-1.56) | 0.834 |
| HAART regimen change | No | 1 | | 1 | |
| | Yes | 1.83(0.86-3.89) | 0.116 | 1.14 (081-2.11) | 0.412 |
| OTC use in the previous 4 weeks | No | 1 | | 1 | |
| | Yes | 1.72 (1.01-2.92) | 0.044 | 1.32 (0.47-2.89) | 0.651 |
| DDI | No | 1 | | | |
| | Yes | 3.41 (2.01-16.67) | 0.003 | 5.66 (1.32-24.18) | **0.019** |
| Herbal use in the previous 4 weeks | No | 1 | | 1 | |
| | Yes | 1.62(0.83-3.15) | 0.158 | 1.03 (0.39-2.40) | 0.514 |

**\*COR** – Crude Odds Ratio, **AOR** - Adjusted Odds Ratio.

of therapy, and usually mild and transient in most patients [45]. Clinicians should be vigilant in early detection and management of these specific ADRs through comprehensive patient assessments and personalized treatment adjustments.

The severity of ADRs reported in this study was mild in 78.3% of all cases. This could be explained partly by the efforts to improve pharmacovigilance and management of side effects in HIV clinics [46–48], most participants did not have any comorbidities, which also increase the severity of ADRs [49,50]. The current drug regimen that most of the participants were taking also included newer ARVs such as dolutegravir-based regimens that have a better toxicity profile, which could explain the mild cases observed [51,52]. This study also found out that about 76.6% of all ADR cases reported were preventable. These findings underscore the importance of pharmaceutical care and therapeutic drug monitoring, including avoiding drugs that had previously caused an ADR, avoiding inappropriate medications, optimizing the dosage regimen, conducting the necessary monitoring tests regularly, and checking for significant drug interactions, all of which have been shown to prevent the occurrence of ADRs among HIV patients on HAART [13,28,53].

Participant's education level and current CD4 count were identified as independent risk factors of ADR occurrence among HIV patients in this study. Participants who had primary and secondary education had about 3.27 and 3.64 odds of developing an ADR respectively, as compared to patients who attended tertiary education. These findings are

comparable to those reported in Ethiopia [54], Northern Nigeria [55], and India [56]. This could be explained by the fact that patients with higher education may understand better drug instructions, adherence, and good nutritional status and promote better self-care and monitoring [57]. Low CD4 count was significantly associated with the occurrence of ADRs, as most patients that developed ADRs had CD4<200 cells/mm3. This is in line with several studies from India [58], Brazil [40] and Ethiopia [33]. Increased risk soon after HAART initiation is thought to result from the rapid rise of CD4 lymphocytes, which can be an important stimulus of ADRs as part of an immune reconstitution inflammatory syndrome [59]. Immunodeficiency can lead to dysregulation in drug metabolism, hence potentially affecting the pharmacokinetics of antiretroviral drugs and increasing the risk of ADRs [60]. Tailored educational interventions focusing on patients with lower educational backgrounds such as simplified medication guides and counselling, to enhance their understanding of drug regimens while promoting adherence. Additionally, patients with low CD4 counts should receive targeted support such as close monitoring, especially during the early stages of HAART initiation.

In this study, the patients with significant DDIs were 5.66 times more likely to develop an ADR and this association was statistically significant. DDIs are more prevalent in HIV patients, and they are important causes of preventable ADRs, as they are at an increased risk of multimorbidity and polypharmacy [61]. DDIs are usually predictable and manageable; there ADRs caused by DDIs may be prevented by monitoring the patient closely or replacing the responsible drugs with other medications in case a DDI is suspected [62]. It is important that healthcare providers regularly conduct comprehensive medication reviews to recognize potentially interacting drugs and intervene in time.

## Limitations of the study

The study's reliance on a single-centre setting could affect the external validity of the study. Secondly, the fact that ADRs were assessed based on patient self-reporting, which might introduce recall bias or underreporting. The study's duration may not capture long-term ADR patterns and the evolving nature of patient responses to HAART. The study also did not capture the adherence level to HAART for the participants, as this could have affected the occurrence of ADRs.

## Conclusion

The study revealed a high prevalence of ADRs (76.6) among HIV patients on HAART at Mbarara Regional Referral Hospital, emphasizing the need for vigilant monitoring and management strategies within HIV care. Education level and current CD4 count demonstrated a significant association with the development of ADRs, highlighting the importance of tailored interventions for these subgroups. The study identified a statistically significant association between clinically significant DDIs and the occurrence of ADRs, underscoring the importance of pharmaceutical care for prompt identification of drug interactions and ADR management in HIV patient care. Future researchers should consider multi-centered and longitudinal studies to generate more precise and generalizable results.

## Acknowledgments

We would like to thank all the staff of ISS clinic of MRRH for their collaboration and support during the data collection. We would also like to extend our thanks to the participants for consenting to take part in this study.

## Author contributions

**Conceptualization:** Nangosya Moses, Narayana Goruntla, Muyindike Rhoda Winnie, Tadele Mekuriya Yadesa.

**Data curation:** Nangosya Moses, Sarad Pawar Naik Bukke, Narayana Goruntla, Bontu Aschale Abebe, Fredrick Atwiine, Muyindike Rhoda Winnie, Tadele Mekuriya Yadesa.

**Formal analysis:** Nangosya Moses, Daniel Chans Mwandah, Fredrick Atwiine, Muyindike Rhoda Winnie, Tadele Mekuriya Yadesa.

**Funding acquisition:** Nangosya Moses, Sarad Pawar Naik Bukke, Daniel Chans Mwandah, Tadele Mekuriya Yadesa.

**Investigation:** Nangosya Moses, Sarad Pawar Naik Bukke, Narayana Goruntla, Bontu Aschale Abebe, Tadele Mekuriya Yadesa.

**Methodology:** Nangosya Moses, Narayana Goruntla, Tadele Mekuriya Yadesa.

**Project administration:** Nangosya Moses, Daniel Chans Mwandah, Tadele Mekuriya Yadesa.

**Resources:** Nangosya Moses, Sarad Pawar Naik Bukke, Narayana Goruntla, Daniel Chans Mwandah, Fredrick Atwiine, Muyindike Rhoda Winnie, Tadele Mekuriya Yadesa.

**Software:** Nangosya Moses, Sarad Pawar Naik Bukke, Bontu Aschale Abebe, Fredrick Atwiine, Tadele Mekuriya Yadesa.

**Supervision:** Nangosya Moses, Muyindike Rhoda Winnie, Tadele Mekuriya Yadesa.

**Validation:** Nangosya Moses, Sarad Pawar Naik Bukke, Daniel Chans Mwandah, Bontu Aschale Abebe, Fredrick Atwiine, Tadele Mekuriya Yadesa.

**Visualization:** Nangosya Moses, Narayana Goruntla, Bontu Aschale Abebe, Muyindike Rhoda Winnie, Tadele Mekuriya Yadesa.

**Writing – original draft:** Nangosya Moses, Fredrick Atwiine.

**Writing – review & editing:** Sarad Pawar Naik Bukke, Narayana Goruntla, Daniel Chans Mwandah, Bontu Aschale Abebe, Muyindike Rhoda Winnie, Tadele Mekuriya Yadesa.

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
