## [Decision Letter · Decision Letter 0]

24 Oct 2024

PONE-D-24-42042Prevalence and Factors Associated with Adverse Drug Reactions Among Patients on Highly Active Antiretroviral Therapy at a Tertiary Hospital in South Western Uganda: A cross-sectional studyPLOS ONE

Dear Dr. Yadesa,

Thank you for submitting your manuscript to PLOS ONE. After careful consideration, we feel that it has merit but does not fully meet PLOS ONE’s publication criteria as it currently stands. Therefore, we invite you to submit a revised version of the manuscript that addresses the points raised during the review process.

We look forward to receiving your revised manuscript.

Kind regards,

Ebenezer Wiafe, PhD, MPharm, Pharm D

Academic Editor

PLOS ONE

Journal Requirements:

3. Please ensure that you refer to Figure 2 in your text as, if accepted, production will need this reference to link the reader to the figure.

Reviewers' comments:

Reviewer's Responses to Questions

**Comments to the Author**

1. Is the manuscript technically sound, and do the data support the conclusions?

Reviewer #1: Yes

Reviewer #2: Yes

Reviewer #3: Partly

2. Has the statistical analysis been performed appropriately and rigorously?

Reviewer #1: Yes

Reviewer #2: Yes

Reviewer #3: No

3. Have the authors made all data underlying the findings in their manuscript fully available?

Reviewer #1: Yes

Reviewer #2: Yes

Reviewer #3: Yes

4. Is the manuscript presented in an intelligible fashion and written in standard English?

Reviewer #1: Yes

Reviewer #2: Yes

Reviewer #3: Yes

5. Review Comments to the Author

Reviewer #1: The topic is interesting and the paper is quite well written. The paper addresses an important issue especially in low income countries where HIV/AIDS is still an issue. The paper is suitable for publication after the revisions have been effected. The rationale is clear and the references used are mostly recent and the references used are mostly recent. The manuscript is well-organized, clearly written and easy to follow. A few typos and grammatical errors can be cleaned up. The methods are scientifically sound and have been described in sufficient detail to address the objectives and allow reproducibility.

Nevertheless, in my opinion, some parts need to be improved, I have some comments:

1. What was your definition of ADR. It would be better to cite it in the method part

2. Since patients were taking a combination of HAART, how do you compute an ADR with a specific medication?

3. In the method section, it would be better if you could explain how you differentiated the ADRs from a disease symptom or if the ADR happens from an ADR of another medication the patient was taking at that point.

4. Table 4: Commonly Reported ADRs among patients with HIV on HAART at MRRH (doesn’t have the frequency)

5. Most of the ADRs look like a subject (patient) reported and It cannot be confirmed. So how did you judge if it is an ADR.

6. How did you characterize some of the ADRs like ‘difficulty of sleeping’ … how many hours of sleep or how do you define this ADR? The same works for Low appetite, diarrhea, excessive eating, drinking…… and so on. (Better to define them in the methods)

7. Since you only had a onetime interaction with the patients (Cross-sectional study), How did you determine the renal problems? Was it based on the data they had in the past?

8. The Discussion could comment more on implications of findings for practice and policy, like what the different stakeholders can do to improve the situation.

Reviewer #2: Title: The title succinctly captures the nature of the work.

Background: The background introduces appropriately the context of the research problem.

Methodology: The cross-sectional design is correct and the tool used, the Naranjo ADR Assessment form is appropriate for the study. The HAART was defined, but the auditors did not indicate the adherence level of these Persons living with HIV. Also, they did not account for other medications that the PLWHIV could be on, be it multivitamins or prophylaxis for Opportunistic infections for instance.

Reviewer #3: The is a very important study that touches on a very important area in HAART. There are some corrections which I have indicated in the attached file for the authors to look at. This is very important to help improve the manuscript

6. PLOS authors have the option to publish the peer review history of their article (what does this mean? ). If published, this will include your full peer review and any attached files.

**Do you want your identity to be public for this peer review?** For information about this choice, including consent withdrawal, please see our Privacy Policy .

Reviewer #1: No

Reviewer #2: No

Reviewer #3: No

---

## [Author Response · Author response to Decision Letter 1]

9 Nov 2024

Response to Reviewers

Reviewer Comment Response

Reviewer 1

1. What was your definition of ADR. It would be better to cite it in the method part The definition has been included and cited (Page 6)

2. Since patients were taking a combination of HAART, how do you compute an ADR with a specific medication? The reported ADR by the participants were evaluated against known adverse reaction profile of each drug using Ugandan Clinical Guidelines 2023 and British National Formulary (BNP) version 85; This was done in order to attribute the reported ADR to specific medication. (Page 6)

3. In the method section, it would be better if you could explain how, you differentiated the ADRs from a disease symptom or if the ADR happens from an ADR of another medication the patient was taking at that point Naranjo ADR assessment scale was used which is a standard tool used determine the causal relationship. The scale helps to determine if the adverse reaction observed is due to the suspected drug or another source be it a disease or other medication (Page 6)

4. Table 4: Commonly Reported ADRs among patients with HIV on HAART at MRRH (doesn’t have the frequency) Frequency has been included in the table

5. Most of the ADRs look like a subject (patient) reported and it cannot be confirmed. So how did you judge if it is an ADR. The Naranjo ADR assessment scale is comprehensive, a standard questionnaire which assigns probability scores, it among other things examines whether the ADR reported started before the initiation of the drug or not. This does not necessarily confirm the ADR but rather provides a probability score in the range of “Doubtful – Possible – Probable – Definite”. As reported under the result section (Pages 13, 14 & Figure 3). The limitation of a possible recall bias due to patient reports was acknowledged under “Limitations of the study” section (Page 19)

6. How did you characterize some of the ADRs like ‘difficulty of sleeping’ … how many hours of sleep or how do you define this ADR? The same works for Low appetite, diarrhea, excessive eating, drinking…… and so on. (Better to define them in the methods) Since ADRs were not predetermined they could not be defined in advance under methods; although all ADRs identified were assessed and reported based on their standard definitions and assessment criteria. Some of the ADRs have been defined under Table 4 (Page 15)

7. Since you only had a onetime interaction with the patients (Cross-sectional study), How did you determine the renal problems? Was it based on the data they had in the past? Objective data on renal function tests was obtained from the reviewed patient charts. An abnormal renal function test result was assessed based on Naranjo scale and known ADR profile of the HAART drugs to determine the causal relationship.

8. The Discussion could comment more on implications of findings for practice and policy, like what the different stakeholders can do to improve the situation. The discussion section has been improved to include specific recommendations for different stakeholders. More recommendations are summarized under the “Conclusion section”.

Reviewer 2 The HAART was defined, but the auditors did not indicate the adherence level of these Persons living with HIV This has been acknowledged under limitations of the study

Also, they did not account for other medications that PLWHIV could be on, be it multivitamins or prophylaxis for Opportunistic infections for instance This was put into account as indicated in Table 5: Other medications including OTCs and herbal medicines use, were evaluated for association with development of ADR

Reviewer 3 There are some corrections which I have indicated in the attached file for the authors to look at. The comments have been attended to accordingly throughout the document.

---

## [Decision Letter · Decision Letter 1]

3 Jan 2025

PONE-D-24-42042R1Prevalence and Factors Associated with Adverse Drug Reactions Among Patients on Highly Active Antiretroviral Therapy at a Tertiary Hospital in South Western Uganda: A cross-sectional studyPLOS ONE

Dear Dr. Yadesa,

Thank you for submitting your manuscript to PLOS ONE. After careful consideration, we feel that it has merit but does not fully meet PLOS ONE’s publication criteria as it currently stands. Therefore, we invite you to submit a revised version of the manuscript that addresses the points raised during the review process.

We look forward to receiving your revised manuscript.

Kind regards,

Ebenezer Wiafe, PhD, MPharm, Pharm D

Academic Editor

PLOS ONE

Reviewers' comments:

Reviewer's Responses to Questions

**Comments to the Author**

1. If the authors have adequately addressed your comments raised in a previous round of review and you feel that this manuscript is now acceptable for publication, you may indicate that here to bypass the “Comments to the Author” section, enter your conflict of interest statement in the “Confidential to Editor” section, and submit your "Accept" recommendation.

Reviewer #1: All comments have been addressed

Reviewer #3: (No Response)

2. Is the manuscript technically sound, and do the data support the conclusions?

Reviewer #1: Yes

Reviewer #3: No

3. Has the statistical analysis been performed appropriately and rigorously?

Reviewer #1: Yes

Reviewer #3: No

4. Have the authors made all data underlying the findings in their manuscript fully available?

Reviewer #1: Yes

Reviewer #3: Yes

5. Is the manuscript presented in an intelligible fashion and written in standard English?

Reviewer #1: Yes

Reviewer #3: Yes

6. Review Comments to the Author

Reviewer #1: (No Response)

Reviewer #3: I raised quite a number of concerns in the first review. However, NONE of the concerns raised were addressed. I think the authors will have to take a look at it again. I did attach a file in PDF with sticky notes on what corrections needs to be addressed. I have therefore attached it again for the concerns to be addressed.

7. PLOS authors have the option to publish the peer review history of their article (what does this mean? ). If published, this will include your full peer review and any attached files.

**Do you want your identity to be public for this peer review?** For information about this choice, including consent withdrawal, please see our Privacy Policy .

Reviewer #1: No

Reviewer #3: **Yes: ** Christian Obirikorang

---

## [Author Response · Author response to Decision Letter 2]

17 Jan 2025

Response to Reviewers

Reviewer Comment Response

Reviewer 3 I raised quite a number of concerns in the first review. However, NONE of the concerns raised were addressed. I think the authors will have to take a look at it again. I did attach a file in PDF with sticky notes on what corrections needs to be addressed. I have therefore attached it again for the concerns to be addressed.

Response: Thank you very much for your crucial comments. The authors believe your comments have helped us to significantly improve our write-up. We have now addressed all the raised comments throughout the document

---

## [Decision Letter · Decision Letter 2]

28 Feb 2025

Prevalence and Factors Associated with Adverse Drug Reactions Among Patients on Highly Active Antiretroviral Therapy at a Tertiary Hospital in South Western Uganda: A cross-sectional study

PONE-D-24-42042R2

Dear Dr. Yadesa,

We’re pleased to inform you that your manuscript has been judged scientifically suitable for publication and will be formally accepted for publication once it meets all outstanding technical requirements.

Kind regards,

Ebenezer Wiafe, PhD, MPharm, Pharm D

Academic Editor

PLOS ONE

Additional Editor Comments (optional):

Reviewers' comments:

Reviewer's Responses to Questions

**Comments to the Author**

1. If the authors have adequately addressed your comments raised in a previous round of review and you feel that this manuscript is now acceptable for publication, you may indicate that here to bypass the “Comments to the Author” section, enter your conflict of interest statement in the “Confidential to Editor” section, and submit your "Accept" recommendation.

Reviewer #1: All comments have been addressed

Reviewer #3: All comments have been addressed

2. Is the manuscript technically sound, and do the data support the conclusions?

Reviewer #1: Yes

Reviewer #3: Yes

3. Has the statistical analysis been performed appropriately and rigorously?

Reviewer #1: Yes

Reviewer #3: Yes

4. Have the authors made all data underlying the findings in their manuscript fully available?

Reviewer #1: Yes

Reviewer #3: Yes

5. Is the manuscript presented in an intelligible fashion and written in standard English?

Reviewer #1: Yes

Reviewer #3: Yes

6. Review Comments to the Author

Reviewer #1: (No Response)

Reviewer #3: Having gone through the revised manuscript, I can say that all my concerns have been addressed and the manuscript is acceptable in the current state.

7. PLOS authors have the option to publish the peer review history of their article (what does this mean? ). If published, this will include your full peer review and any attached files.

**Do you want your identity to be public for this peer review?** For information about this choice, including consent withdrawal, please see our Privacy Policy .

Reviewer #1: No

Reviewer #3: **Yes:** Prof Christian Obirikorang

---

## [Editor Report · Acceptance letter]

PONE-D-24-42042R2

PLOS ONE

Dear Dr. Yadesa,

I'm pleased to inform you that your manuscript has been deemed suitable for publication in PLOS ONE. Congratulations! Your manuscript is now being handed over to our production team.

Kind regards,

on behalf of

Dr. Ebenezer Wiafe

Academic Editor

PLOS ONE